# A chiral molecular propeller designed for unidirectional rotations on a surface

Yuan Zhang[1], Jan Patrick Calupitan [2,3,4], Tomas Rojas[5,6], Ryan Tumbleson[1,6], Guillaume Erbland[4], Claire Kammerer [4], Tolulope Michael Ajayi[1,6], Shaoze Wang[1,6], Larry A. Curtiss[5], Anh T. Ngo[5], Sergio E. Ulloa [6], Gwénaël Rapenne[2,3,4] & Saw Wai Hla [1,6]

Synthetic molecular machines designed to operate on materials surfaces can convert energy into motion and they may be useful to incorporate into solid state devices. Here, we develop and characterize a multi-component molecular propeller that enables unidirectional rotations on a material surface when energized. Our propeller is composed of a rotator with three molecular blades linked via a ruthenium atom to a ratchet-shaped molecular gear. Upon adsorption on a gold crystal surface, the two dimensional nature of the surface breaks the symmetry and left or right tilting of the molecular gear-teeth induces chirality. The molecular gear dictates the rotational direction of the propellers and step-wise rotations can be induced by applying an electric field or using inelastic tunneling electrons from a scanning tunneling microscope tip. By means of scanning tunneling microscope manipulation and imaging, the rotation steps of individual molecular propellers are directly visualized, which confirms the unidirectional rotations of both left and right handed molecular propellers into clockwise and anticlockwise directions respectively.

[1] Center for Nanoscale Materials, Argonne National Laboratory, Lemont, IL 60439, USA. [2] Division of Materials Science, Nara Institute of Science and Technology, NAIST, 8916-5 Takayama-cho, Ikoma, Nara 630-0192, Japan. [3] NAIST-CEMES International Collaborative Laboratory for Supraphotoactive Systems, Toulouse, France. [4] CEMES, Université de Toulouse, CNRS, 31013 Toulouse, France. [5] Materials Science Division, Argonne National Laboratory, Lemont, IL 60439, USA. [6] Nanoscale and Quantum Phenomena Institute and the Department of Physics and Astronomy, Ohio University, Athens, OH 45701, USA. Correspondence and requests for materials should be addressed to G.R. (email: gwenael-rapenne@ms.naist.jp) or to S.W.H. (email: hla@ohio.edu)

Synthetic molecular machines are fascinating and have a great promise to revolutionize scientific and technological fields. The immense interest on this research area is evident by the 2016 Nobel Prize in Chemistry awarded for the "design and synthesis of molecular machines". To date, various molecular machines that can be operated on solid surfaces such as molecular vehicles and molecular motors have been developed[1–7]. Among them, molecular propellers have shapes similar to their macroscopic counterparts and they are formed by molecular scale blades that rotate along a shaft[8]. In nature, molecular propellers are vital in many biological applications ranging from the swimming of bacteria to intracellular transport[9,10] but, unlike bio-molecular propellers, synthetic molecular propellers may be able to operate in much harsher environments[11,12]. However, the molecular machines can change their mechanical, and electronic properties upon adsorption on a surface, which can greatly impact their performance. In some cases, the surface can also impose chirality in the molecule as well[13]. Therefore, for designing molecular machines to be operated on surfaces, the effects of the surface should be considered, and if possible should be used as an advantage. Although there are a number of molecular systems having propeller shapes and functions that can be deposited on a solid surface[14], further development of molecular machines demands exquisite and robust design of molecular propellers with precise control over their rotation direction.

Here, we develop a multi-component molecular propeller that enables unidirectional rotation suitable to operate on a material surface. An important aspect of molecular machine development is to test the properties and operations of individual molecular machines. We use a variety of scanning tunneling microscope (STM) tip manipulation schemes; electric field induced rotation, inelastic electron tunneling (IET) induced rotation, and rotation by mechanical force to investigate unidirectional rotation of the molecular propellers on one-molecule-at-a-time basis. In electric field induced rotations, the propeller rotates by using the electrical energy supplied from the STM tip. Here, the propeller can be rotated by the tip scanning with a high voltage above ±1 V during acquisition of images, as well as by positioning the tip statically above or beside the molecule. The threshold electric field required for the rotation, 0.25 V/Å, is determined from the slope of the linear relationship between the negative threshold voltage and the tip height. Using this value, the electrical energy stored in the molecular propeller is calculated as −0.66 eV. While a low current is used for the electric field induced rotation, IET induced rotations are performed with a high current above 2 nA at positive biases. The IET induced rotation is initiated by the transfer of electron energy from inelastic tunneling electrons via temporary electron attachment to the lowest unoccupied molecular orbital (LUMO) of the molecular propeller. The threshold energy for the IET induced rotation is determined as ~0.6 eV. Furthermore, mechanical rotations of both left and right handed propellers provide information on the detailed rotation mechanism, which reveals that swinging of the propeller blades occur before proceeding to a full rotation. Moreover, we are able to directly visualize the rotation steps of the individual propellers from the STM images acquired at each rotation steps on both left and right handed geometries. The statistical analyses gleaned from the STM images and mechanical rotation events further confirm the unidirectional rotation of the propeller. The experimental results are corroborated by density functional theory calculations, which reveal tilting of the stator phenyl rings that leads to the formation of a ratchet shape molecular gear.

## Results

**Molecular propeller structure**. The rotor blades of our molecular propeller are composed of three indazole groups that are arranged in a chelating tripodal manner and bind to a ruthenium (Ru) atom in a facial mode while retaining thioester (SEt) groups at opposite ends (Fig. 1a, b). The stator is composed of a cyclopentadienyl center (Cp) with five p-bromophenylene (Cp(PhBr)$_5$) units covalently attached to it[15,16]. The Ru atom is coordinated to both the stator and to the three propeller blades (Fig. 1a). For experiments, the molecular propellers are deposited onto an atomically clean Au(111) surface using a custom built Knudsen cell under an ultrahigh vacuum (UHV) environment.

On a Au(111) surface, large area STM images acquired at 80 K substrate temperature show individual molecular propellers preferentially located along the wider face centered cubic (FCC) regions of the herringbone surface reconstructions (Fig. 1c). The molecular propellers adsorb on the Au(111) surface by positioning the Cp unit of the Cp(PhBr)$_5$ stator parallel to the surface plane. In our previous study[5], the three propeller blades formed by the indazole groups have been used as a tripodal stator to anchor the molecular motor by preferentially binding the sulfur atoms from the three SEt groups on a Au(111) surface. In the current system however, the five bromophenyl groups of the Cp(PhBr)$_5$ stator are preferentially positioned on a Au(111) surface and the propeller blades formed by the three indazole groups are located at the top with an angle of 120 degrees to each other (Fig. 1b, d). Remarkably, at 80 K substrate temperature, some of the propellers appear as round shape structures in STM images indicating that these propellers are rotating (Fig. 1c, e). Thus, the molecules here not only have the shape of a propeller but also have the propeller function. When the same sample is cooled down to 5 K substrate temperature, such round shape structures are no longer present on the surface, and all the molecular propellers appear stationary. Therefore, the propeller rotation observed at 80 K is initiated by thermal excitation.

Interestingly, the STM images of the molecular propellers reveal left or right handed chirality depending on the positioning of the three propeller blades (Fig. 2a). Statistical analysis acquired from 500 molecular propellers show approximately equal probability of the two enantiomers as expected (Fig. 2b). This finding is in contrast to molecular propellers in solution under ambient conditions where chirality is not observed because the two atropisomers, which exist via twisting of the tripodal ligand, are in equilibrium with a very fast exchange rate.

To understand the observed molecular propeller structure on Au(111) surface, we have performed density functional theory (DFT) calculations. Geometrically relaxed calculations reveal that the Cp ring of the stator is located 3.7 Å above the top surface layer of Au(111) (Fig. 2c), and the binding of the stator to the Au(111) surface stabilizes the tilting of the phenyl-rings. Due to the steric repulsions between adjacent phenyl rings, all the phenyl-rings are tilted to the same direction[17,18], either left or right. Here, the five phenyl-rings are tilted with different angles; 18°, 35°, 43.4°, 66.5°, and 76.7° respectively from the surface plane (Fig. 2d, and Supplementary Fig. 1). The different tilting angles here are induced by the interaction of the phenyl-rings with the indazole rings forming the propeller blades located above (Fig. 2e), and they are found to be not dependent on the adsorption site due to a weak molecule-surface binding. To confirm this further, we have calculated the structure of the stator without the propeller blades and the Ru atom. Unlike the complete propeller structure, the stator-only calculations give the tilt angle of phenyl rings as ~39° from the surface plane (Supplementary Fig. 2). The DFT calculation further reveals that rotation of the propeller blade alters the tilt angle of the phenyl rings. Here, when the propeller blade is approached towards the phenyl ring, its tilt angle becomes smaller (Supplementary Movie 1). Such orientation of the five phenyl-rings dictates the skewing direction of the upper indazole rings thereby producing the observed chirality. As a

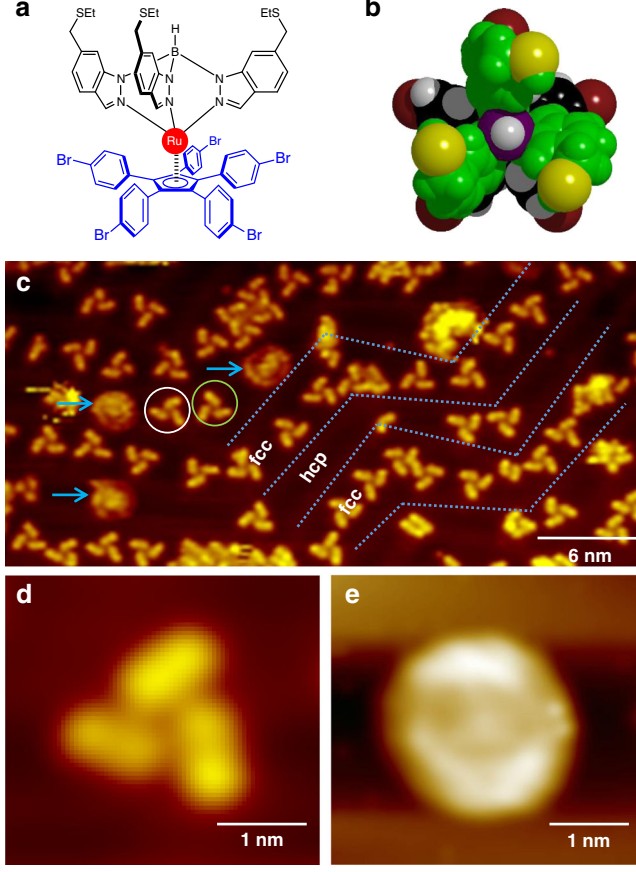

**Fig. 1** Shape and structure of the molecular propeller. **a** Chemical structure of the molecule. **b** Top view of the ball and stick model of the molecule. color code: Dark red = Br, gray = H, black = C atoms on the Cp(PhBr)$_5$, violet = B, yellow = S, green = other atoms on the indazole fragments (ethyl groups are omitted for clarity). **c** A large area STM image of molecular propellers on Au(111) at 80 K. Tripod structures are stationary. Blue arrows point to rotating propellers. Gray dashed lines highlight the herringbone structure of Au(111). STM images showing a stationary (**d**) and rotating molecular propeller (**e**). [STM imaging parameters for **c**, **d**, and **e**: $V_t = -1$ V, $I_t = 70$ pA]

result, the indazole rings are arranged in a pseudo-helical conformation, similar to a macroscopic propeller, through the axis going from the Ru to the B atoms, and through the center of the Cp ring of the molecule (Fig. 2e). The strain that induces this chirality is mostly due to the steric hindrance of adjacent phenyl rings and interaction with the surface[13], although the molecule-surface binding is weak. The directional tilting of the five phenyl-rings in the stator here is vital because it now acts as a ratchet-shaped molecular gear (Fig. 2f). Owing to the symmetry breaking by the surface and adjacent steric repulsion[19], switching the tilt direction will induce a higher energy barrier. This means that the propeller blades should rotate towards the same direction as the tilting of the phenyl-rings. Therefore, the stator acting as a ratchet-shape molecular gear here should impose a unidirectional rotation of the propeller.

**Propeller rotation**. Indeed, stepwise rotation of the individual molecular propellers can be observed when the STM images are acquired with biases equal or greater than ±1 V. Fig. 3a, b present an example of such rotation. Figure 3a shows two right-handed and two left-handed molecular propellers on Au(111) surface imaged with −0.5 V bias. When the same area is imaged again

with −1.0 V bias, one of the right-handed (marked with 'P') and one of the left-handed (marked with 'M') propellers are rotated stepwise to anticlockwise and clockwise directions, respectively (Fig. 3b), while the other two propellers remain static. The stepwise rotation events can be captured during the imaging as well. Figure 3c shows an initial position of a right-handed molecular propeller. In the next consecutive scanned image shown in Fig. 3d, the propeller appears as already rotated to the anticlockwise direction in the upper part of the image. During the STM tip scanning at −1.0 V, another stepwise rotation occurs triggering an abrupt change in contrast (indicated with an arrow in Fig. 3d). Following this event, the rest of the propeller is imaged as 48° rotated from the initial position (Fig. 3e).

Since the three propeller blades are positioned on the 5-teeth ratchet gear formed by the stator there are 15 steps to complete a 360° rotation. This means that each step has a 24° rotation, and the propeller shown in Fig. 3c–e has been rotated to anticlockwise direction for two full steps. The indazole groups that form the propeller blades can swing from ±5° to ±30°. For instance, the left-handed propeller 'M' in Fig. 3a is rotated a full clockwise step, however only two blades of the right-handed propeller 'P' are rotated to full anticlockwise steps while the third blade appears only a slightly tilted angle from its original position. Thus this propeller does not achieve a full rotation. Geometrically relaxed DFT calculations reveal that such swinging of the propeller blade has a modest energy cost, from 45 meV to 68 meV, depending on the rotating angle and the location of the blade with respect to the phenyl rings of the stator (Supplementary Fig. 3). At 80 K substrate temperature where the experiments are performed, the propeller in such intermediate state may overcome the rotation barrier with the help of thermal excitation and may initiate rotation.

The threshold bias required to rotate the propellers can be determined from the statistical analysis gleaned from the STM images acquired at different biases (Fig. 3f). In order to avoid the rotation caused by the mechanical contact with the tip, a low tunneling current (therefore a large tip height) from 10 to 70 pA is used for all the image acquisitions. Here, the rotational angles are determined by taking an average of all three propeller blades. Because of a slight swinging of the propeller blades occur frequently, the average angles do not appear as multiple of 24°. From the plot of rotation angle as a function of bias, the threshold bias required to rotate the molecular propellers is found as ±1 V (Fig. 3f).

To check the rotational directions of both left and right-handed propellers, a statistical analysis for the rotations observed during STM imaging is performed. Here, only full step rotations of all three propeller blades are counted and from a total of 168 rotation events of both left and right-handed propellers, 132 rotations (78.6%) are observed as correct rotation directions, i.e., left-handed propellers rotate to the clockwise direction while the right-handed propellers rotate to the anticlockwise direction, respectively (Fig. 3g). Among them, 116 rotations are the single step rotations while only 16 are observed in the double step rotations. For the reverse rotations, i.e., left and right-handed propellers rotate to anticlockwise and clockwise directions, respectively, only single step rotations are observed. Because the observed rotations mainly lead into single rotation steps, we can conclude that the preferential rotation of the molecular propeller is unidirectional.

**Controlled electric field and inelastic tunneling electron induced rotations**. The observed rotations during image acquisition invoke further questions concerning the required electric field strength and electrical energy stored in the molecular

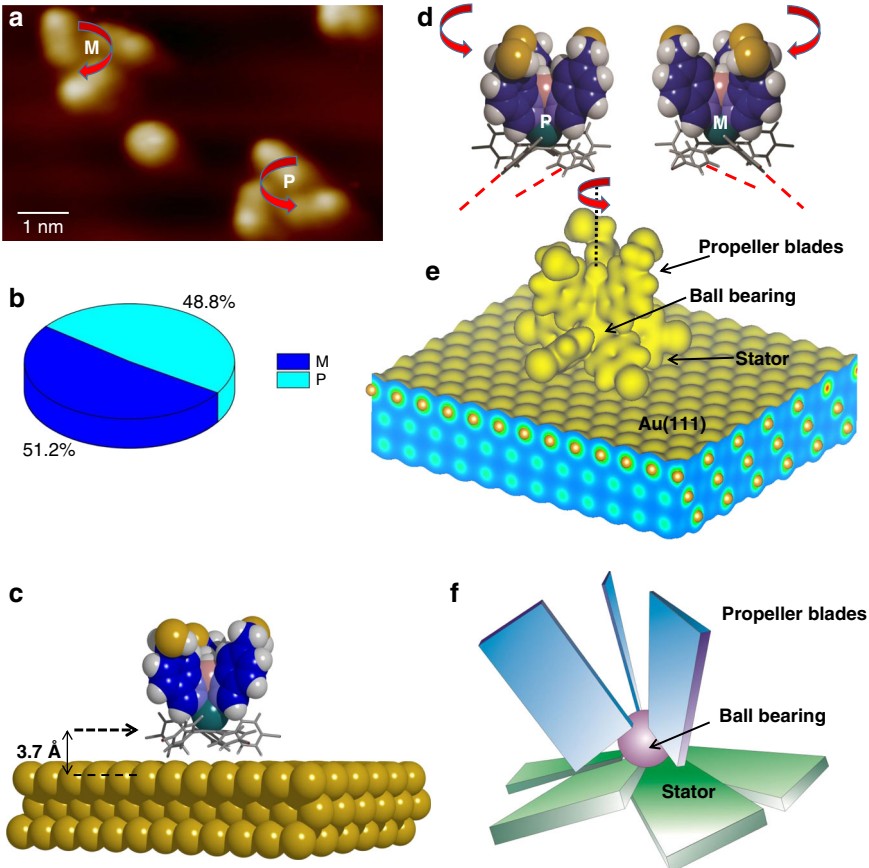

**Fig. 2** Chirality and electronic structure. **a** STM image of two molecular propellers on Au(111) surface showing left-handed (M) and right-handed (P) chirality [Imaging parameters: $V_t = -1.2$ V, $I_t = 70$ pA]. **b** The pie-chart shows the statistics of M and P. **c** DFT calculated structure of a molecular propeller adsorbed on Au(111) slab (the ethyl groups on the indazole rings are omitted). The arrow indicates the Cp plane of the stator unit. **d** The molecular propeller structures for 'M' and 'P' are induced by the left and right tilting of the π-rings of the stator. The red-dashed lines indicate different tilting angle of the π-rings. **e** A 3-D electron density plot of the molecule adsorbed on Au(111) surface as revealed by the DFT calculation. **f** A drawing depicting a ratchet type stator that drives unidirectional rotation

propeller. To determine the threshold electric field required for rotation, we perform controlled stepwise rotations of the propeller using STM tip-induced electric field manipulation as follows: Initially, the tip is positioned above the center of the propeller with a fixed height (Fig. 4a), and then the bias is ramped from 0 V to −3.5 V while the corresponding tunneling current is recorded. During this process, a sudden change in tunneling current occurs if the propeller rotates (Fig. 4b). The rotation is confirmed by STM imaging after the process (Fig. 4c). The experiments are repeated by changing the tip height, and to avoid the tunneling electron involvement, a low current with negative bias (from 20 pA to 0.5 nA) is used for such tip-induced rotations. The recorded threshold bias exhibits a linear relationship with tip height (Fig. 4d), and from the slope of the plot the threshold electric field is determined as 0.25 V/Å. Using this electric field value, we have calculated the electrical energy stored in the propeller as −0.66 eV (Supplementary Method 1 and Supplementary Table 1).

The electric field induced rotation can also be realized with the tip positioning next to the propeller. Figure 4e shows the recorded tip positions for such rotation events. Here, the inner circle indicates the perimeter of the propeller while the outer circle, which has 1 nm larger radius than the inner circle, is drawn for eye guidance. Like in previous case, the propeller is rotated by ramping the tunneling bias from 0 to −3.5 V. Rotation occurs if the lateral distance of electric field source, i.e., the tip, is less than

1 nm from the perimeter of the propeller. Since the tip is not positioned directly above the propeller here, tunneling current involvement is minimized, and the electric field is the primary source of the rotation.

The propeller can also be rotated by inelastic electron tunneling (IET) processes[20,21]. To trigger the IET process, the probability of electron capture by the molecule is important and a sufficient tunneling current is necessary. Unlike the electric field induced rotation where a low tunneling current (pA regime) and a high bias are used, the IET induced rotations are realized with a high current (greater then 2 nA) with a lower positive tunneling voltage of 0.6 V. The use of positive bias means that the electrons are tunneling from the tip to the unoccupied states of the sample in our case. Figures 4f–h present a sample IET induced rotation. Here, the STM tip is positioned directly above the propeller at a fixed height with the tip set-point current of 2 nA (Fig. 4f). Then a tunneling bias of 0.6 V is applied for 30 s, and the corresponding tunneling current is recorded as a function of time (Fig. 4g). The changes in the tunneling current can be associated with the rotation events, which can be confirmed by STM imaging after the manipulation (Fig. 4h). Typically, an IET induced rotation is triggered by a transfer of electron energy via a temporary electron attachment to an unoccupied orbital. Figures 4i, j present the calculated highest occupied and lowest unoccupied molecular orbitals (HOMO and LUMO) of the propeller adsorbed on Au (111) surface. The LUMO occupies the central Ru atom, as well as

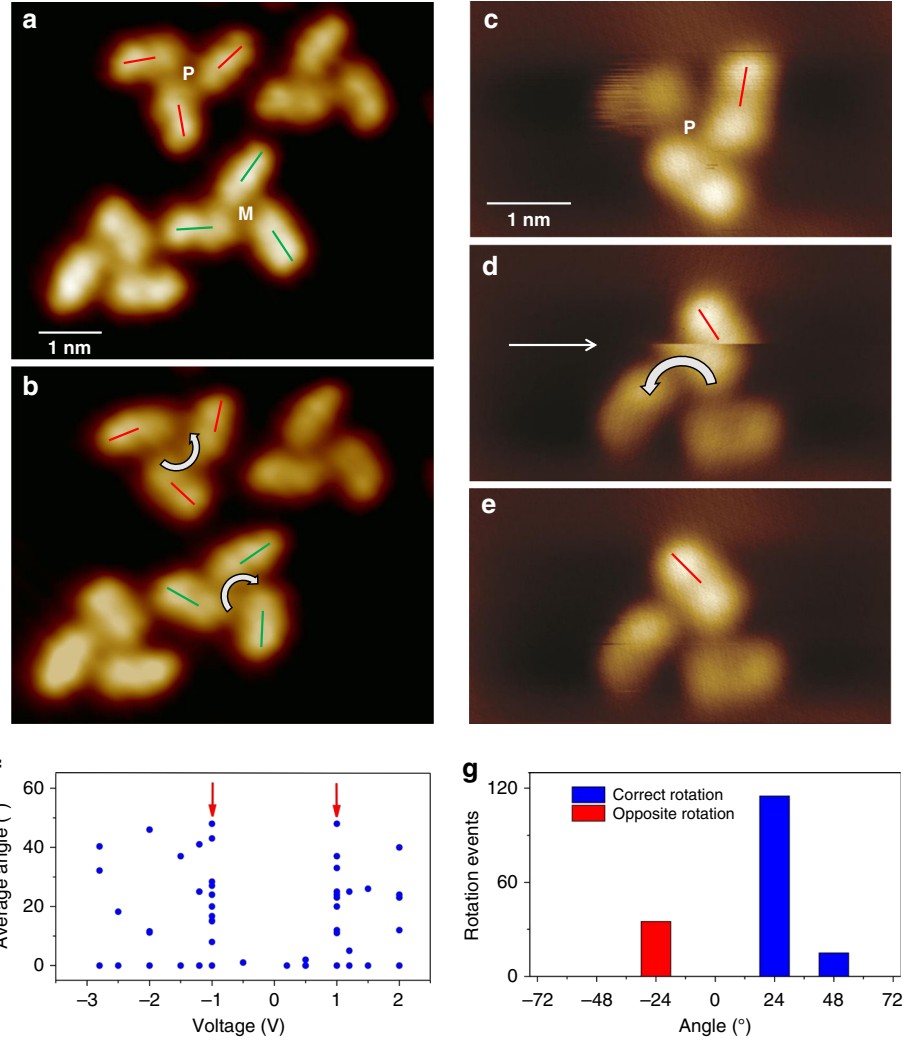

**Fig. 3** Unidirectional rotation. **a** STM image of molecular propellers on Au(111) surface where the respective chirality of two of the propellers are indicated [Imaging parameters: $V_t = -0.5\,V$, $I_t = 17\,pA$]. **b** When the same area was scanned with $-1\,V$, the 'P' and 'M' propellers are rotated to anticlockwise and clockwise directions, respectively [Imaging parameters: $V_t = -1.0\,V$, $I_t = 17\,pA$]. **c** An STM image of a 'P' propeller. **d** During a consecutive imaging, the propeller is rotated causing a sudden change in the image contrast (indicated with an arrow). **e** The final position of the propeller after the rotations [Imaging parameters: $V_t = -1.0\,V$, $I_t = 50\,pA$]. **f** Average rotation angle as a function of bias. The arrows indicate the threshold bias at $\pm 1\,V$. **g** A statistical plot of the rotation

the phenyl rings of the stator, and the vibrational relaxation through the excitation of this orbital could result in the observed rotation[5]. This can be directly confirmed by a threshold energy measurement, which shows that ~0.6 V is necessary to trigger the IET manipulation while the dI-dV spectroscopy and the projected density of states unveil this energy as close to the energy of the LUMO orbital (Supplementary Fig. 4). Therefore we attribute the observed IET rotation as triggered by a temporary electron attachment to the LUMO orbital of the propeller (Supplementary Discussion 1).

**Mechanical rotation**. Next, we examine the rotation mechanism of both enantiomers by mechanically rotating the propeller blades with the STM tip. The mechanical rotations are performed by using tunneling resistances from 1.3 MΩ to 70 MΩ, and with the bias range of 0.01–0.1 V. For these rotations, the tip is initially positioned above the Au(111) surface next to a propeller blade and then the tip height is reduced for 3 Å to 4 Å from its initial position. Then the STM tip is laterally moved towards the propeller blade in a constant current mode, i.e., maintaining the tip

height by the electronic feedback loop, until it mechanically pushes the blade[22] (Fig. 5a). During this process, the tip-height trace is recorded as a function of distance. A successful rotation can be confirmed by comparing the STM images acquired before and after the manipulation. The recorded mechanical rotation signals show a small dip proceeding a pushing like curve[22,23] as the examples presented in Fig. 5b. These mechanical rotation signals can be explained as follows (Fig. 5c): When the tip approaches the propeller blade (1), it swings away from the tip until it is stuck with the up-ramp of the molecular gear tooth formed by a tilted phenyl ring (2). This swinging motion of the blade causes the tip height to reduce thereby producing a dip (2). Next, the tip climbs up the propeller blade whereby the lateral push-force of the tip increases (3). When the push force exceeds the threshold lateral force to overcome the barrier, the propeller is rotated stepwise. Since the propeller blade is now moved away from its position underneath the tip, the tip height suddenly drops to maintain the constant current by the electronic feedback loop (4). The latter part is similar to the pushing of individual atoms on a surface with an STM tip[22,23] except that the propeller blade here is rotated on the ratchet shape molecular gear. The

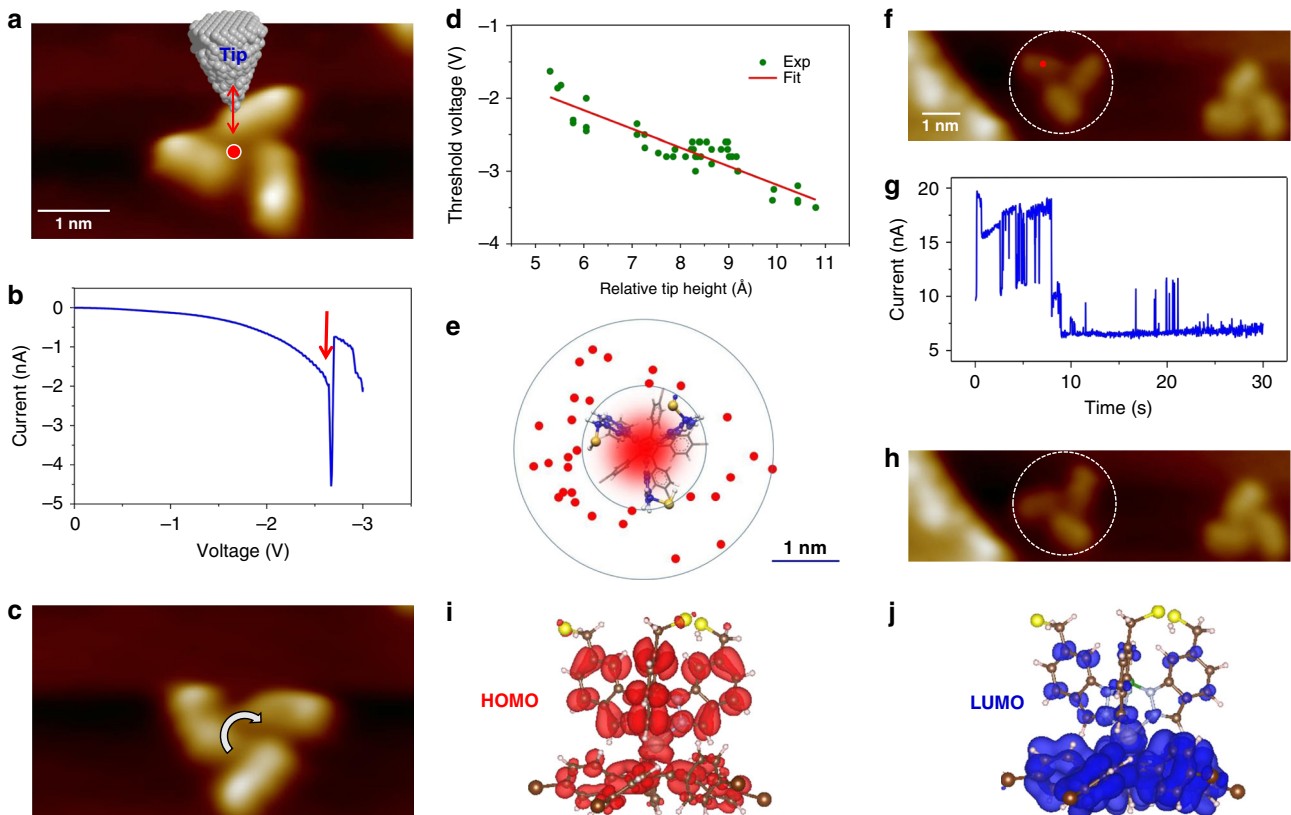

**Fig. 4** STM tip-induced rotations. **a** The tip is positioned above the center of a propeller (red dot) for a threshold electric field measurement. The red arrow indicates variation of tip heights in successive measurements. **b** The I–V plot shows a sudden change of current indicated with a red arrow, from which the threshold bias for rotation is determined. **c** STM image after the manipulation shows the rotated propeller. [Tip set point parameters for rotation: $V_t = -2.0$ V, $I_t = 4 \times 10^{-10}$ A. Imaging parameters: $5.4 \times 3.0$ nm$^2$ area, $V_t = -2.0$ V, $I_t = 75$ pA]. **d** Threshold voltage has a linear relationship with tip height. **e** The red dots indicate the tip positions for the threshold voltage measurement at the side of the molecule. **f** The tip is positioned above the red dot for IET induced rotation. **g** The abrupt changes in the current are caused by the movement of the propeller blades. **h** The propeller is rotated anti-clockwise after the IET manipulation. [Tip set point parameters for rotation: $V_t = 0.6$ V, $I_t = 2$ nA. Imaging parameters: $10.4 \times 3$ nm$^2$ area, $V_t = -2.0$ V, $I_t = 46$ pA]. **i** Calculated HOMO and **j** LUMO orbitals of the propeller

successful rotation of the propeller is then confirmed by comparing the STM images acquired before and after the mechanical rotation.

From the mechanical rotation experiments, one can unravel the rotation mechanism as swinging the blade followed by passing the molecular gear teeth for each rotation step. Such mechanical manipulation could result in rotating just one propeller blade or the whole propeller. For instance, STM images of Fig. 5d, e show the initial and final stages of a mechanical rotation where just one propeller blade is rotated. The rotation of a single blade here requires only a small energy, from 45 meV to 68 meV (Supplementary Fig. 3). A sequence of STM images presented in Fig. 5f–h shows stepwise rotation of a left-handed molecular propeller into clockwise direction. Similarly, Fig. 5i, j, and 5k present the stepwise mechanical rotations of a right-handed molecular propeller. Because of the tri-blade propeller symmetry, the initial structure of the propeller is reproduced in every 5 rotation steps, i.e., in 120°. The rotation of both left and right handed chiral propeller can be directly visualized by animating the STM images acquired at the each rotation step (Supplementary Movie 2). Figure 5l presents the number of mechanical rotation events as a function of the average rotation angle of the three propeller blades. Since this process is taking into account swinging angle of the propeller blades, the average rotation angles are not confined to the multiple of 24°. However, the Gaussian fit

of the statistical data clearly unveils the peak rotation angle as 24° (Fig. 5l), and thus a full step rotation is preferred.

The STM tip can be used to forcefully rotate the propeller blades into the reverse direction as well. The process is similar to the one described above but the tip pushing direction now is opposite to the preferred rotation direction, i.e., clockwise rotation for the right-handed and anticlockwise rotation for the left-handed propellers (Supplementary Fig. 5). The propellers can be rotated stepwise into the reversed directions using similar manipulation parameters as above and the Gaussian fit from the statistical analysis (Fig. 5m) gives the peak rotation angle as 18.5°. This means that most of the reverse rotations are not a full step rotation, i.e., 24° or higher angles. Moreover, the mechanical manipulation signals for the reverse rotation process shows much larger pushing curves than the rotation into the correct directions indicating a larger push force[22,23] (Supplementary Fig. 5). This finding indicates that although a forceful manipulation of the propellers into a reverse direction is possible, it is not a preferred direction for the rotation. This is in good agreement with the electric field and IET induced rotations of the propellers.

In summary, we have developed a multi-component molecular propeller that can be operated on materials surfaces when energized. Unidirectional rotations of the molecular propeller are demonstrated on a gold crystal surface using STM tip manipulation schemes on one-propeller-at-a-time basis. Unidirectional

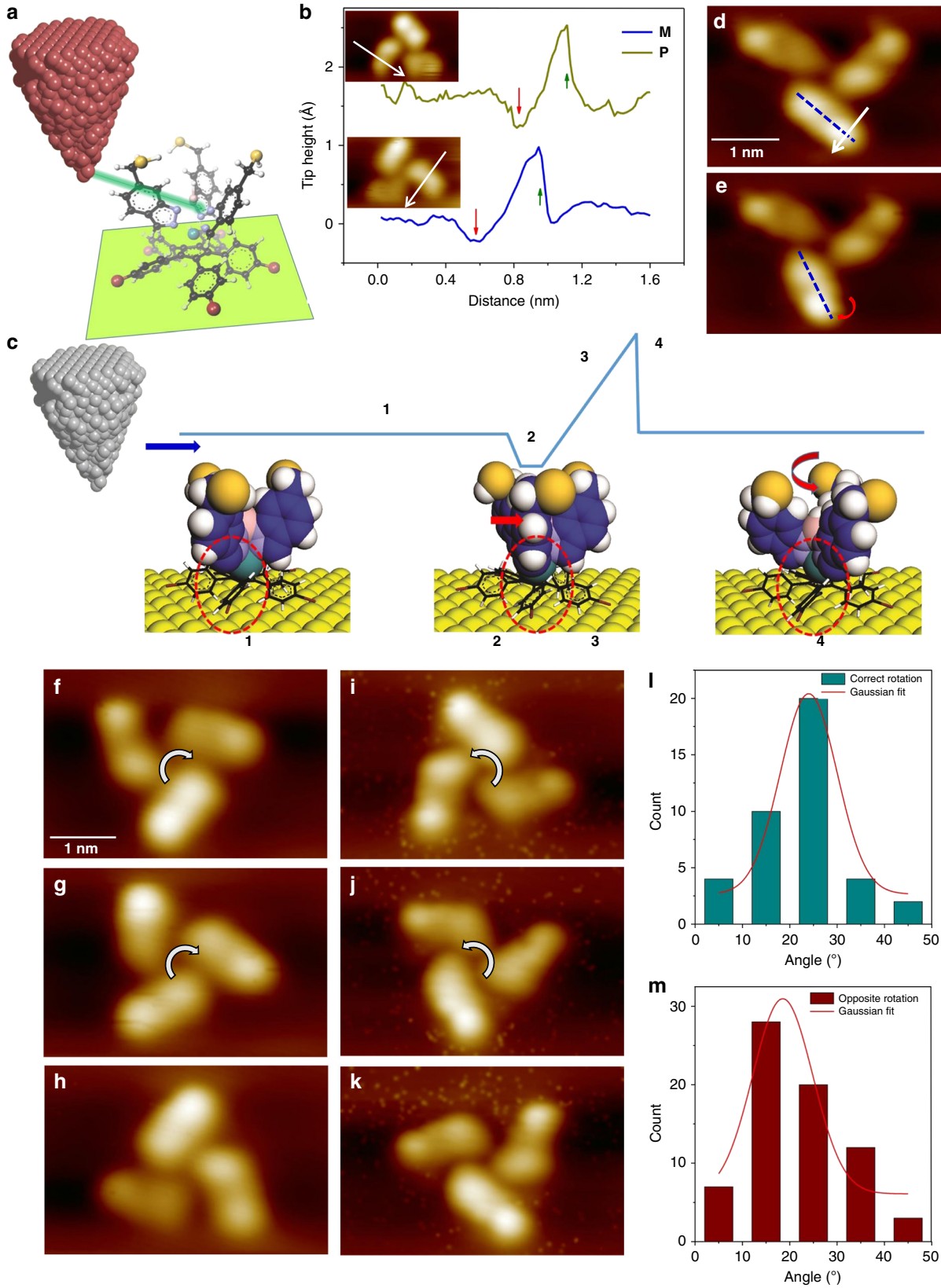

rotation here is dictated by the stator forming a ratchet shape gear that leads to dynamical chirality in the propellers. An important aspect to discuss here is whether they are useful to actual work. We find that a full rotation of the blades can displace other molecules located next to the propeller indicating that they can be used to move a molecular load (Supplementary Fig. 6). For some applications, it may be useful to consider cascading the propellers. However, our molecular propellers may not be suited for such cascade-gear applications because of their trigonal geometry with approximately 120° angle between the blades. Such a large angle

**Fig. 5** Propeller rotation by mechanical manipulation. **a** A drawing depicting the mechanical rotation. **b** Mechanical manipulation signals during a single step rotation of M and P propellers. Inset: The arrows indicate the path and direction of manipulations. The top manipulation signal is vertically displaced 1.5 Å for clarity. **c** The mechanical rotation process. The dashed circles indicate the propeller bottom and the position of a phenyl ring underneath. **d** STM image before and **e** after a single propeller blade rotation. White arrow in **d** indicates the manipulation direction [Imaging parameters: $V_t = -0.1$ V, $I_t = 50$ pA]. **f** An initial STM image of a left-handed molecular propeller. **g** and **h** After single step rotations into clockwise direction. **i** STM image of a right-handed propeller and **j** and **k** are after stepwise rotations into anticlockwise directions. [Imaging parameters: $V_t = -0.1$ V, $I_t = 50$ pA] **l** and **m** Rotation angle as a function of count for the forward and reverse rotations, respectively

could result in slippage and thus the molecular gear with more teeth would be required[24]. A key demonstration of our work is that the substrate surface can be used to engineer chirality of the molecular propellers and thus not only the internal structure of the molecules but also the substrate should be considered for the design of the molecular machines to be operated on solid surfaces. For instance, it could be envisioned that using a particularly patterned substrate, the molecular propeller having mono-chirality may be able to form selectively for potential applications. Of great interest for chiroptical devices and asymmetric synthesis, propeller chirality has been mostly studied theoretically and experimentally in solution. The design and operation of a chiral propeller motor formed by multiple components at the single molecule level on a surface here will have impact not only for potential applications in a solid state environment but also for the further development of complex molecular machines and their operations.

## Methods

**Sample and tip preparation**. The single crystal Au(111) sample was cleaned by a repeated sequence of $Ar^+$ ion sputtering and annealing. Before deposition of the molecules, the cleanliness of the sample was checked by imaging. The molecular compound was synthesized as follows:[15,16] Initially, pentaphenylcyclopentadiene is first brominated on the para-positions of the phenyl and on the cyclopentadiene core using neat bromine giving an hexabrominated compound. After subsequent oxidative addition on the ruthenium carbonyl cluster, conversion of piano-stool complex into the molecular propeller was finally achieved by reaction with the potassium salt of thioether-functionalized hydrotris(indazolyl)borate in presence of thalium sulfate. The overall yield for 3 steps was 63%. The molecules were deposited onto atomically clean Au(111) sample using a custom built Knudsen source under an ultrahigh vacuum (UHV) environment. For the deposition, the molecular source was heated to ~450 K and the Au(111) substrate temperature was held at ~120 K. The sample was then transferred to a Createc GmbH STM scanner directly attached to the sample preparation chamber via a gate valve under UHV condition. The sample temperature was then lowered to 80 K and 5 K for respective experiments. For all the experiments, an electrochemically etched tungsten wire was used as the STM tip. The tip was prepared in-situ by gently dipping into the substrate prior to the manipulation experiments[25] and thus it was assumed to cover with Au.

**STM imaging and tip-induced propeller rotations**. The STM tip induced electric field, inelastic tunneling electrons, and the mechanical rotations of the propellers were performed at 80 K substrate temperature. STM imaging was performed at 80 K and 5 K substrate temperatures, and a low tunneling current range between 10 pA and 70 pA was used. The mechanical rotations of the both chiral propellers with the STM tip were performed by using tunneling resistance range of 1.3 MΩ to 70 MΩ, the tunneling current range of 16 nA to 75 nA, and the bias range of 0.01 V to 0.1 V. The initial set point tunneling current and voltage before the manipulation were 50 pA, and ±1 V, respectively.

**DFT theory calculations**. Spin polarized Density Functional Theory (DFT) calculations were carried out with the Vienna ab initio simulation package code[26], with core electrons described by the projected augmented wave method[27]. Exchange-correlation was treated in the Generalized Gradient Approximation[28]. Because of the relative importance of non-bonding molecule surface interactions, van der Waals D3 functional was used[29]. The plane wave basis was expanded to a cutoff of 600 eV. The Au(111) surface was modeled by a three-layer slab containing 432 atoms and the first Brillouin zone was sampled at the Γ point only. The molecular propeller composed of 117 atoms was placed on top of a three-layer Au (111) slab representing the surface with a vacuum space of 20 Å. The geometry optimizations were converged within 2 meV per formula unit for the total energies.

## Data availability

All data needed to evaluate the conclusions in the paper are present in the paper and/or the Supplementary Information. Additional data related to this paper may be requested from the authors.

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

## Acknowledgements

All the experimental characterizations in this work were supported by the US Department of Energy, Office of Basic Energy Sciences grant DE-FG02-02ER46012. Molecule synthesis was supported by the ANR ACTION project ANR 15-CE29-0005, CNRS, the University Paul Sabatier (Toulouse), and the JSPS KAKENHI grant in aid for Scientific Research on Innovative Areas "Molecular Engine (No.8006)" 18H05419. Use of the Center for Nanoscale Materials, an Office of Science user facility, was supported by the U.S. Department of Energy, Office of Science, Office of Basic Energy Sciences, under Contract No. DE-AC02-06CH11357. Work by L.A.C. was supported by the US Department of Energy, BES Materials Sciences under Contract DE-AC02-06CH11357. J.P.C. thanks the NAIST International Scholarship for his PhD funding and the NEXT Labex for his mobility grant (Programme Investissements d'Avenir ANR-11-IDEX-0002-02, and reference ANR-10-LABX-0037-NEX). We gratefully acknowledge the computer time from the Argonne National Laboratory Computing Resource Center (LCRC).

## Author contributions

S.W.H. conceived and designed the experiments: Y.Z., J.P.C, R.T., T.M.J., and S.W. performed the STM experiments; J.P.C. and G.E. synthesized the molecular propeller, C.K. and G.R. designed the molecule, T.R. and A.T.N. performed the calculations, L.A.C. and S.E.U. guided the calculations, Y.Z., J.P.C., R.T., T.M.J., S.W., and S.W.H. analyzed the experimental data. All the authors discussed the results and commented on the paper.

## Additional information

**Competing interests:** The authors declare no competing interests.

