## [Peer Review File · Nature Communications]

Reviewers' Comments:

Reviewer #1:

Remarks to the Author:

The paper describes the manipulation of a single-molecule rotor by the tip of an STM. Whereas in solution, the molecular blades can rotate in both directions, adsorption on a surface breaks the symmetry of the stator, thus inducing a ratchet-like configuration with a well-defined directional sense of the rotation.

The crucial requirement for inducing the directionality of the rotation is the adsorption configuration of the stator on the surface. The inclination angles of the stator wings have been derived from DFT simulations. The resulting angles differ each individual phenyl ring with one of them being almost perpendicular to the surface (76 degree). As there is no experimental evidence for this configuration, the calculations need to be described in more detail. How do the inclination angles vary with adsorption site? Is the described configuration the only energetic minimum in the calculations? Are there other possible configurations, which may explain that some rotors are freely rotating at 80 K while the others are not?

Authors observe a voltage threshold of rotation of +/- 1 V. Is this independent of the tunneling current? They state that the mechanism is electric-field induced? Did they check the dependence of rotation threshold with tip distance? Are there any molecular orbitals that may mediate the rotation, similarly to the observation of some of the authors in a similar rotor (Ref. 5)?

The authors state that the "threshold energy" is 1V. Do they mean 1 eV? Assuming an electric-field induced mechanism, they should calculate the energy stored in the electric field.

In some of the manipulation events only one of the blades is swinging. Does this require the same energy? Since the adsorbed molecule is claimed to be highly asymmetric (see inclination angles discussed above), one may ask if the asymmetry leads to different rotation properties (threshold energy, barrier height) for different tip positions.

Without more detailed explanations and quantitative insights, I cannot recommend publication in Nature Communications.

Reviewer #2:

Remarks to the Author:

This is a nice publication where the authors demonstrate the unidirectional rotation of a molecular propeller on a gold crystal. They used electrical energy provided by the scanning tunneling microscope tip and demonstrated to induce step-wise rotation of the propeller managing in the same time to directly visualize the rotation steps of the individual propellers for both geometries left and right handed. Besides that, they have also studied the rotation mechanism in more detail. By using forceful manipulation with the tip they demonstrate that reversible rotation is possible however not in the preferred rotation of the propellers. While the study is rigorous and clearly demonstrate the finding it is less clear how such finding could be useful or of interest to the wider community. Several questions would be important to address. What is the influence of the density of motor on the surface and how do they move when an obstacle is on the way? Can such molecular motors show coordination and simultaneous switching when present in close vicinity?

On line 111 the authors mention that "The strain that induces this chirality (of the molecular motor) is due not only to the steric hindrance of adjacent phenyl rings, but also to the interaction with the surface as well. In that case how the authors envision that they can influence the chirality by engineering the surface?"

In conclusion the manuscript is interesting for the Nat Commun audience after appropriate revision if they can address the questions raised above.

Reviewer #1 (Remarks to the Author):

The paper describes the manipulation of a single-molecule rotor by the tip of an STM. Whereas in solution, the molecular blades can rotate in both directions, adsorption on a surface breaks the symmetry of the stator, thus inducing a ratchet-like configuration with a well-defined directional sense of the rotation. The crucial requirement for inducing the directionality of the rotation is the adsorption configuration of the stator on the surface.

ANS: We thank the referee for carefully reading the manuscript and providing criticisms that allow us to improve our manuscript. We have performed additional experiments and DFT calculations to address the referee's questions, and 4 new authors, 2 theorists, and 2 experimentalists are added for the new theory and experiments. Please find our detail responses below.

The inclination angles of the stator wings have been derived from DFT simulations. The resulting angles differ each individual phenyl ring with one of them being almost perpendicular to the surface (76 degree). As there is no experimental evidence for this configuration, the calculations need to be described in more detail. How do the inclination angles vary with adsorption site?

ANS: The inclination angles of the stator's phenyl rings are induced by the propeller (rotator) arms (please see the supplementary movie S1). To check this further, we have calculated the structure of the stator only and the result does not show such tilt angle variation (supplementary info S2).

The following sentences are added in p6, line109: *"To confirm this further, we have calculated the structure of the stator without the propeller blades and the Ru atom. Unlike the complete propeller structure, the stator only calculations give the tilt angle of phenyl rings as $\sim 39^\circ$ from the surface plane (Supplementary Information S2). The DFT calculation further reveals that rotation of the propeller blade alters the tilt angle of the phenyl rings. Here, when the propeller blade is approached towards the phenyl ring, its tilt angle becomes smaller (See Supplementary Movie S1)."*

We have checked the geometry of the propeller by moving half of the surface atom site (adsorption site) but the total energy and structure remains the same because it is weakly bind to the surface (physisorbed).

The following sentence is added in page 6, line 107: *"and they are not dependent on the adsorption site due to a weak molecule-surface binding."*

Is the described configuration the only energetic minimum in the calculations? Are there other possible configurations, which may explain that some rotors are freely rotating at 80 K while the others are not?

The geometrically relaxed DFT calculations give the described configuration as the minimum energy configuration. However, there are intermediate states that are closed to the minima. For instance, rotation of only one arm could lead to an intermediate state with only slightly higher energy (from 45 meV to 68 meV) depending on the blade position with respect to the pi-ring (added in Supplementary Info S3). The following sentences are added in page 7, line 145: *"Geometrically relaxed DFT calculations reveal that such swinging of the propeller blade has a modest energy cost, from 45 meV to 68 meV, depending on the rotating angle and the location of the blade with respect to the phenyl rings of the"*

stator (Supplementary Information S3). At 80 K substrate temperature, the propeller in such intermediate state may overcome the rotation barrier with the help of thermal excitation and may initiate rotation.”

At 80K, the molecule with a slightly different configuration with an energy close to the minimum energy could be unstable and it may trigger rotation by thermal excitation.

Authors observe a voltage threshold of rotation of +/- 1 V. Is this independent of the tunneling current? They state that the mechanism is electric-field induced? Did they check the dependence of rotation threshold with tip distance? Are there any molecular orbitals that may mediate the rotation, similarly to the observation of some of the authors in a similar rotor (Ref. 5)?

ANS: To answer the referee's questions, we have performed additional experiments and a new result section: "Controlled electric field and inelastic electron tunneling induced rotations" is added in page 9, and page 10. A new figure (Fig. 4) is also added.

In particular, we have measured tip height dependent threshold bias (New figure: Fig. 4a to 4d). The threshold voltage for rotation, $\pm 1V$, was initially determined from the STM images where rotations occurred during scanning. The rotation here is induced by the electric field because we use very low tunneling current (in pA range) and high bias ($\pm 1V$ or higher) for image acquisition. Moreover, the rotation can occur when the tip is not directly above the propeller (New figure: Fig. 4e), which clearly indicates the electric field induced rotation without involving the tunneling current at low tunneling regime.

In addition, we can rotate the propeller by IET process (added new figure: Fig. 4f to 4h) using high tunneling current above 2nA with voltage as low as 0.6V. This process should involve the LUMO orbital. We have also added the calculated HOMO and LUMO orbital shapes in the new figure (Fig. 4i, and 4j). The corresponding texts can be found in the new section added in page 9 and 10.

The authors state that the "threshold energy" is 1V. Do they mean 1 eV? Assuming an electric-field induced mechanism, they should calculate the energy stored in the electric field.

ANS: Thank you for pointing out our error. In page 1, line 28; page 9, line 162, and page 9, line 168: We have replaced the word "energy" with "bias". Using the determined threshold electric field of 0.25V/Å for rotation, we have calculated the energy stored in the molecular propeller as -0.66 eV (added in supplementary information S4).

The following sentence is added in page 10, line 191: "*Using this field value, we have calculated the electrical energy stored in the propeller as -0.66 eV (Supplementary Information S4).*"

In some of the manipulation events only one of the blades is swinging. Does this require the same energy? Since the adsorbed molecule is claimed to be highly asymmetric (see inclination angles discussed above), one may ask if the asymmetry leads to different rotation properties (threshold energy, barrier height) for different tip positions.

ANS: We have calculated the energy difference between the minimum energy and that for one blade rotated configuration as 0.045 eV to 0.068 eV depending on the rotated blade position with respect to the stator phenyl ring. This energy range is about 10% or less as compared to the threshold electric field

energy of 0.66 eV. As discussed above the asymmetry of the phenyl ring tilt angles are dependent on the position of the blade, i.e. when the blade is closer to the phenyl ring, and it changes tilt angle while the ring away from the blade also changes to a larger angle to compensate this change. As a result, the net energy barrier for rotation remains constant.

Without more detailed explanations and quantitative insights, I cannot recommend publication in Nature Communications.

We have answered all of the referee's questions and we believe that the new calculations and experiments greatly improve our manuscript.

Reviewer #2 (Remarks to the Author):

This is a nice publication where the authors demonstrate the unidirectional rotation of a molecular propeller on a gold crystal. They used electrical energy provided by the scanning tunneling microscope tip and demonstrated to induce step-wise rotation of the propeller managing in the same time to directly visualize the rotation steps of the individual propellers for both geometries left and right handed. Besides that, they have also studied the rotation mechanism in more detail. By using forceful manipulation with the tip they demonstrate that reversible rotation is possible however not in the preferred rotation of the propellers.

ANS: We thank the referee for carefully reading the manuscript and providing criticisms that allow us to improve our manuscript. Please find our detailed responses below. We have also performed new experiments and theoretical calculations.

While the study is rigorous and clearly demonstrate the finding it is less clear how such finding could be useful or of interest to the wider community. Several questions would be important to address. What is the influence of the density of motor on the surface and how do they move when an obstacle is on the way? Can such molecular motors show coordination and simultaneous switching when present in close vicinity?

We have also added a new supplementary section (S7) to demonstrate this. If the molecular propellers are closely packed, however, coordinated rotation is difficult. This is because of the trigonal (3-arms) geometry. A new reference (ref. 27) discussing this aspect is also added.

ANS: The following sentences are added in page 15, line 307: *"An important aspect to discuss here is whether they are useful to actual work, i.e. to remove the cargo. Although the rotator blades can swing, we find that they can be used to remove the molecular load (Supplementary Information S7). Moving the load here is achieved through the steric repulsion between the molecules. However, to use these molecular propellers as cascade-gears in a one dimensional propeller chain is difficult because of their trigonal geometry with approximately 120° angle between the blades. Such a large angle could result in the slippage and thus the molecular gear with more teeth would be required²⁷."*

On line 111 the authors mention that “The strain that induces this chirality (of the molecular motor) is due not only to the steric hindrance of adjacent phenyl rings, but also to the interaction with the surface as well. In that case how the authors envision that they can influence the chirality by engineering the surface?”

ANS: The following sentences are added in page 16, line 314: “An important demonstration of our work is that surface can be used to engineer chirality of the molecular propellers and thus not only the internal structure of the molecules but also the substrate surface should be considered for the design of the molecular machines to be operated on solid surfaces. For instance, it could be envisioned that using a particularly patterned substrate, the molecular propeller having mono-chirality may be able to form selectively for potential applications.”

In conclusion the manuscript is interesting for the Nat Commun audience after appropriate revision if they can address the questions raised above.

We have answered all of the referee’s questions and we believe that the new calculations and experiments greatly improve our manuscript.

Reviewers' Comments:

Reviewer #1:

Remarks to the Author:

The manuscript has been improved substantially compared to the first version. However, a few questions arise and/or remain open.

The manuscript distinguishes between three regimes of molecular rotation mechanisms. It is not clear how they can be distinguished in such a clear manner. The electric field is also present in the regime where rotation is claimed to be of inelastic origin. How large is the field at 0.6 V and 2 nA? Is there a clear threshold energy which can be reconciled with molecular orbitals, thereby justifying the inelastic origin?

A similar question applies to the force-induced manipulation section. How large is the electric field in case of these manipulations? Which types of forces are at play?

Is the orientation of the electric important? Data is only shown for negative sample bias.

Presumably, x axis of Fig. 4d does not state the absolute tip height. Probably it should be a relative tip height.

The y axis in Fig. 5b should be in real units (as in Supplement). The x axis requires an inset of manipulation trace along the molecule in an STM image.

The new paragraph about "removing the cargo" is not understandable without section S7. The wording should be improved for readability without the supplement.

Distinction between different operation regimes is interesting, but needs to be corroborated in more detail. This may help to clarify if there is an advancement in understanding the rotation mechanism as compared to their earlier publication (Ref. 5) in Nature Nanotechnology 2013.

Reviewer #1 (Remarks to the Author):

The manuscript has been improved substantially compared to the first version. However, a few questions arise and/or remain open.

ANS: Thank you for carefully reading the manuscript. Please find our point by point answers below.

The manuscript distinguishes between three regimes of molecular rotation mechanisms. It is not clear how they can be distinguished in such a clear manner.

ANS: Since the STM manipulation techniques and corresponding mechanisms have been developed for well over 20 years, we have overlooked the references. We have added a review article that explains all three STM manipulation techniques as a reference (ref. 21) and an early IET paper (ref 21) in page 11, line 214. References are renumbered. More explanations are also added in Supplementary S5.

The electric field is also present in the regime where rotation is claimed to be of inelastic origin. How large is the field at 0.6 V and 2 nA?

ANS: From the tunneling resistance of 0.3 G Ω (a relative tip height of $\sim 6\text{\AA}$), we estimate an electric field of $\sim 0.1\text{V}/\text{\AA}$, which is less than the threshold electric field of $0.25\text{ V}/\text{\AA}$. Therefore, this field strength is not sufficient to trigger the rotation (discussed in Supplementary Information S5).

Is there a clear threshold energy which can be reconciled with molecular orbitals, thereby justifying the inelastic origin?

ANS: Yes, there is a clear threshold bias. We have added threshold bias measurement plot in Supplementary S5, which clearly reveals the threshold bias as around 0.6 V. This energy value coincides with the LUMO orbital energy from the dI/dV signal (Supplementary S5).

The following paragraph is added in page 12, line 231: *“This can be directly confirmed by a threshold energy measurement, which shows that $\sim 0.6\text{V}$ is necessary to trigger the IET manipulation. The dI-dV spectroscopy measurement and the calculated projected density of states plot (Supplementary Information S5) reveal that this energy is close to the energy of the LUMO orbital of the molecule. Therefore we attribute the observed IET rotation as triggered by a temporary electron attachment to the LUMO orbital of the propeller.”*

A similar question applies to the force-induced manipulation section. How large is the electric field in case of these manipulations? Which types of forces are at play?

ANS: Force induced manipulations were performed with a bias range 0.01 of 0.1 V, and the estimated electric field strength is $0.004\text{V}/\text{\AA}$ to $0.02\text{ V}/\text{\AA}$. The pushing signal clearly indicates that repulsive force via tip-molecule contact is the key (discussed in Supplementary S5).

Is the orientation of the electric important? Data is only shown for negative sample bias.

ANS: Fig. 3f clearly shows that both positive and negative polarity can be used to rotate. The electric field as a function of tip height is only measured for the negative bias. At the positive bias, the IET induced manipulations can occur at higher current above 0.5 nA, and therefore we have not performed tip height dependent measurement.

Presumably, x axis of Fig. 4d does not state the absolute tip height. Probably it should be a relative tip height.

ANS: Referee is correct. We have changed the 'x' axis label as the 'relative tip height' in Fig. 4d.

The y axis in Fig. 5b should be in real units (as in Supplement). The x axis requires an inset of manipulation trace along the molecule in an STM image.

ANS: The real unit is now shown in the 'y' axis in Fig. 5b. Molecule images with the traces of manipulation paths are also added.

The new paragraph about "removing the cargo" is not understandable without section S7. The wording should be improved for readability without the supplement.

ANS: We have rephrased the sentence in page 16, line 323 as, "*We find that a full rotation of the blades can displace other molecules located next to the propeller indicating that they can be used to remove a molecular load*".

Distinction between different operation regimes is interesting, but needs to be corroborated in more detail. This may help to clarify if there is an advancement in understanding the rotation mechanism as compared to their earlier publication (Ref. 5) in Nature Nanotechnology 2013.

ANS: As mentioned above, we have added more clarification to distinguish the manipulation modes in Supplementary S5 and ref. 21.